**Data Availability Statement:** Data has been anonymised and will be held on LSHTM's COMPASS. We have included all transcripts (in-

# Understanding antimicrobial use in subsistence farmers in Chikwawa District Malawi, implications for public awareness campaigns

Eleanor E. MacPherson[1]*, Joanna Reynolds[2], Esnart Sanudi[3], Alexander Nkaombe[3], John Mankhomwa[3], Justin Dixon[4], Clare I. R. Chandler[5]

1 Blantyre Malawi and Department of Clinical Sciences, Malawi-Liverpool-Wellcome Trust, Liverpool School of Tropical Medicine, Liverpool, United Kingdom, 2 Sheffield Hallam University and Capacity Q, Sheffield, United Kingdom, 3 Malawi-Liverpool-Wellcome Trust, Blantyre, Malawi, 4 Department of Global Health and Development, London School of Hygiene and Tropical Medicine, London, United Kingdom, 5 Department of Global Health and Development, London School of Hygiene and Tropical Medicine, London, United Kingdom

* eleanor.macpherson@lstmed.ac.uk

## Abstract

Drug resistant infections are increasing across the world and urgent action is required to preserve current classes of antibiotics. Antibiotic use practices in low-and-middle-income countries have gained international attention, especially as antibiotics are often accessed beyond the formal health system. Public awareness campaigns have gained popularity, often conceptualising antimicrobial resistance (AMR) as a problem of excess, precipitated by irrational behaviour. Insufficient attention has been paid to people's lived experiences of accessing medicines in low-income contexts. In Chikwawa District, Malawi, a place of extreme scarcity, our study aimed to understand the care and medicine use practices of households dependent on subsistence farming. Adopting an anthropological approach, we undertook medicine interviews (100), ethnographic fieldwork (six-month period) and key informant interviews (33) with a range of participants in two villages in rural Chikwawa. The most frequently used drugs were cotrimoxazole and amoxicillin, not considered to be of critical importance to human health. Participants recognised that keeping, sharing, and buying medicines informally was not the "right thing." However, they described using antibiotics and other medicines in these ways due to conditions of extreme precarity, the costs and limitations of seeking formal care in the public sector, and the inevitability of future illness. Our findings emphasise the need in contexts of extreme scarcity to equip policy actors with interventions to address AMR through strengthening health systems, rather than public awareness campaigns that foreground overuse and the dangers of using antibiotics beyond the formal sector.

## Introduction

In November 2018, the phone of one of the authors (EMP) alerted her of a message, it was from the WhatsApp group of a project she was working on. An image popped up that she

depth interviews and structured medicine interviews) from the facility based and community-based components of the study. In line with our ethical approval, we have excluded raw fieldnotes due to the challenges in maintaining confidentiality. The DOI for the data in its repository: https://doi.org/10.17037/DATA.00001771.

**Funding:** This study was funded by the Foreign, Commonwealth & Development Office, United Kingdom (FIEBRE Project PO7856) (EM, JR, ES, AN, JD, CC) and AMR Cross-Council Initiative through a grant from the Medical Research Council MR/S004793/1 (EM, JM, CC). The funders had no role in study design, data collection and analysis, decision to publish, or preparation of the manuscript.

**Competing interests:** The authors have no competing interests to declare.

instantly recognised: it was from a public health campaign in the UK to raise awareness of antimicrobial resistance. She recalled seeing it in her primary health care practice in the UK. It was a picture of a hand reaching into a bowl with sweets mixed in with capsules representing antibiotics. The text accompanying the image warned of the dangers of misusing antibiotics and the need to always speak to a healthcare professional before using them. In Malawi, the image was being shared on WhatsApp, an instant messaging platform as part of the World Antibiotic Awareness Week, a global campaign coordinated by the World Health Organization. She had spent the previous week talking to health care workers and residents about the essential medicine situation in the rural district of Chikwawa. The narratives and stories shared were ones of extreme scarcity with health workers struggling to manage patients in the context of extremely limited resources including antibiotics, and for residents who faced challenging decisions of deciding whether to pay for transport to take a household member to the clinic or skip a meal for the day. The notion that antibiotics were so freely available that people could take them as unthinkingly as sweets seemed to come from another world. Not Chikwawa where scarcity shaped many aspects of people's lives including access to health care. This paper is a response to this uncomfortable juxtaposition; an exploration of how the jarringly distinct antibiotic imaginaries across different spaces become blurred in singular entreaties to protect these medicines.

The catastrophic impact of antimicrobials losing their effectiveness has been well documented, with apocalyptic predictions of medicine returning to the "dark ages" [1,2]. In the past two decades, countries across the globe have reported rises in drug resistant infections [3]. O'Neill predicted that without prompt action, deaths from AMR could increase to 50 million per year by 2050 with the majority of these occurring in low- and middle-income countries (LMICs) [2]. The ramifications of AMR extend more widely to impacts on human development and food security [3]. Policy statements by international agencies, including the WHO-led Global Action Plan to tackle AMR, emphasise the complexity of the problem and the need to address AMR across a One Health spectrum—considering human, animal, and environmental health holistically [4–6]. Action has been slow due to significant funding constraints, particularly in low-income contexts [7].

Academics and policy actors have drawn parallels between climate change and AMR conceptualising them both as "wicked policy problems" [8]. They are considered "wicked" because they are caused by intractable social and environmental issues, are irreversible in nature, require urgent collective action but lack simple planning solutions [9,10]. For both AMR and climate change, the ramifications are being most acutely felt by those in low-income contexts where policy makers and citizens lack the resources to rapidly adapt to reduce the impact. For AMR, weaker health systems and limited access to second-and third-line antibiotic classes makes treating patients with drug resistant infections challenging [11]. For climate change, the inability of precarious urban settlements and rural communities to adapt their housing and livelihood strategies to withstand extreme weather means living conditions are worsening [12,13]. Consensus among global actors and international agencies exists on the urgent need to reduce the dependence on antibiotics and reduce carbon emissions [14,15]. Parallels can be drawn between climate action and AMR in terms of the ways in which in which global policy has focused on individual behavioural change interventions, often overlooking the ways societal structures and economic models also require change [16,17]. In these interventions, everyone is being urged to change their behaviour for the wider global good. Yet, the poorest and most marginalised within societies are the least able to adapt and absorb policy changes [18]. Serious questions need to be posed about whether everyone should be asked to change their behaviour for the wider global good.

In the field of social science research on AMR there is a growing consensus that "off-the-shelf toolkits" to reduce antibiotic may be ineffective in LMICs [4,5,19]. Learning lessons from studying climate change, we can see that there is a need to carefully trace what parts of life are entangled with the phenomenon to find effective solutions [13]. Social research and anthropological studies have generated evidence to inform the types of interventions that might be effective particularly in low income contexts, where governments face significant financial constraints [19–24]. While, social scientists and others have widely appealed to consider the context when developing interventions, there are still few concrete examples of change [25].

## Precarity in Chikwawa, Malawi

The concept of precarity can be conceptualised as an embodied state of vulnerability that reflects the interdependence of bodies on others for survival [26]. Precarity describes a politically induced condition where certain populations suffer from failing social and economic networks of support that creates differential exposure to injury, violence and death [27]. First used to theorise class relations and neoliberal capitalism, it has been extended to conceptualise a general, pervasive ontological condition of vulnerability [28]. In the Chikwawa District of Malawi, a district of Southern Malawi, where we based our study, precarity is a state experienced by many residents. As one of the poorest countries in the world, and ranked 174 out of 189 countries in the Human Development Index [29], urbanisation has been slower in Malawi than many other countries, and approximately eighty percent of population continue to reside in rural locations [30]. As a rural district in a post-colonial state, Chikwawa has historically been overlooked in economic and development policies. During colonial rule, which ended in 1964, land was appropriated by the colonial rulers and distributed to European settlers to grow crops for export creating extractive agricultural policies that continue to shape land ownership and use today [31,32].

Opportunities for formal or wage employment are rare and state welfare provision extremely limited. The district is considered semi-arid, and many households survive on rain-fed, low-input farming to meet the food needs of the household [33]. Those participating in this type of farming are often referred to as subsistence farmers. In Chikwawa, 72% of households are considered to have very low food security and subsistence farmers often struggle to ensure there is sufficient food for their households [34]. Joshua and colleagues' (2016) study in Chikwawa found that climate change, including the failure of perennial rainfall was resulting in declining yields of subsistence farmers crops worsening living conditions [13].

Chikwawa district has a heavy burden of infectious diseases, and one of the highest rates of malaria in Malawi [35–37]. The government provides access to medical care without user fees, but accessing services can often place a high financial burden on households through out-of-pocket expenses such as transport [38]. Substantial geographical inequalities exist with uneven health care coverage leaving rural areas underserved particularly when accessing District and Tertiary level health services [39,40]. As described elsewhere, we have previously shown that in primary health care facilities in Chikwawa, care centred on the provision of an antimicrobial. Amid chronic lack of essential medicines and other resources, health workers relied on only a few medicines–with cotrimoxazole and lumefantrine/artemisinin (antimalarial) drugs provided by international donors being the most frequently prescribed and dispensed to patients. Clinical interactions were tightly scripted which provided patients with little time to ask questions or negotiate their treatment. Fever was conceptualised by participants as more than just a symptom of malaria, reflecting global health campaign messages that not all fever is malaria [41].

The regulatory framework governing access to antibiotics and antimalarials prohibits purchase without a prescription. In practice, antibiotics and antimalarials can be bought from

private clinics, pharmacies, drug shops, grocery and from informal drug vendors without a prescription [42]. Crackdowns on community-based drug vendors are reported regularly in the national newspapers [43]. The aim of our study was to provide an in-depth understanding of the ways medicines were accessed and used by households dependent on subsistence farming in the Chikwawa District of Malawi. We selected subsistence farmers in a low-income context because they are a group often economically marginalised yet still targeted with images and messages of awareness raising campaigns developed in the Global North.

## Materials and methods

This study used an ethnographic methodology (S1 Text) that was informed by the principles of critical medical anthropology, which emphasises the need to attend to the social, economic and political context of ill-health and healing [44]. The study was conducted as part of the FIEBRE study, a multi-country and multidisciplinary investigation of febrile illness and antimicrobial use in five countries in Africa and Asia [45]. The social science study in Malawi explored the emergent roles of antimicrobials in primary health care settings and in residential areas in Chikwawa. This paper reports the findings from the research with residents. The findings link to the research carried out in health facilities which are presented elsewhere [41]. The study used a range of qualitative ethnographic methods (described below) to answer three questions [46]: (1) what are the most frequently used antibiotics in Chikwawa district? (2) How do subsistence farmers in rural Chikwawa seek medicines, care, and recover from illness? (3) How do antimicrobials and other medicines circulate in formal and informal settings in rural Chikwawa? The study ran from January 2019 until March 2020. The fieldwork was carried out by researchers (EM, AN, ES and JM) employed by the Malawi-Liverpool-Wellcome Trust, an international research organisation, with a prominent and established presence in the region.

We began the fieldwork with structured medicine interviews in 100 households, purposively sampling 16 villages from across the district to ensure a wide geographical variation in our sample. We used a "drug bag" during the interviews; this was a bag of drugs we assembled from locally available antibiotics from both formal and informal sources [47]. Using the drug bag, we took participants through a series of cumulative pile sorting exercises, to establish recognised and frequently used antibiotics (S2 Text). By using the physical presence of the medicines, participants did not require knowledge of the names of the drugs but instead where able to recognise by sight. We were then able to use the medicines to explore how the drugs were sought and used in the household and any challenges participants encountered with access.

In May 2019, we suspended data collection as life-threatening flooding forced 86,976 people into displacement camps [48]. When fieldwork restarted (6 weeks later), we carefully selected two sites for in-depth ethnographic work that had been least affected by the floods. The residential areas selected, were also the catchment areas for the health centres we sampled for the health care study, providing the opportunity for rich insights into health care seeking and care practices and allowing for triangulation across the two data collection sites. The ethnographic fieldwork was led by ES and AN, both Malawian researchers with experience carrying out ethnographic and qualitative research in the region, over a 6-month period in two villages. As a woman, ES found herself interacting more with female participants and, as a man, AN with men. We used a purposive sampling approach, aiming to include men and women living and working within the two villages that were selected based on relationships established during the structured medicine interviews. ES and AN stayed in the villages for extended periods, including overnight. Day-to-day they spent time in households and in communal spaces to develop a picture of livelihood and leisure activities. The broad focus of the study was to understand how people and households within the village responded to ill-health, and how

medicines were accessed and used in this response. During initial community entry activities, this focus was emphasised during village meetings. AN and ES used the networks they established in the two villages to follow up with any household where a family member was sick, visiting the compound to understand the illness episode. They also spent time with the village health committee and at a health post based in one of the villages. Notes were hand-written at the end of each day in field diaries and then typed up and shared between the team at the end of each week.

To explore themes arising in further depth, a total of 33 interviews were conducted with residents in the same villages, purposively sampling men and women AN and ES had previously interacted with. All interviews were recorded, transcribed, and translated into English from Chichewa by the research team.

Frequencies of antibiotic recognition and use in households were established based on the structured medicine interviews. Analysis of the data collected from the ethnographic fieldwork and in-depth interviews was conducted through an inductive, iterative analytic approach, drawing primarily on a thematic analysis approach [44]. To ensure all team members contributed to the analysis, during the six-month data collection period, we held weekly debriefing sessions with the team to reflect on data collection progress and processes. During these meetings we identified and addressed any ethical concerns encountered in the field and identified emerging themes and any new insights or avenues for enquiry to be followed up in the next week. We also built on work initially undertaken within the primary health care facilities. The development of the first coding frame took place iteratively, building on the reviewing of the data, and the debriefing sessions. All transcripts and fieldnotes were imported into NVivo 12 and coded line-by-line to group ideas and develop themes to explain emergent phenomena. To complete the analysis two further analysis workshops held virtually with the whole co-authorship team to allow for and reflection throughout the process.

Ethical approval was obtained from the College of Medicine Research Ethics Committee Malawi (P06/182429) and London School of Hygiene and Tropical Medicine Research Ethics Committee (14617). Permission to work in the district was provided by the district health authorities, and in all villages, approval was provided by the Chief or their representative. Before the ethnographic research began in the villages, community meetings were held to inform people of the study and approval was sought from local governance structures. For households included in the ethnography and in-depth interviews, written or witnessed thumb prints consent was provided.

## Results

Three broad themes reflect our analysis of the social, economic, and moral relations that shape how and why people seek medicines outside the formal health sector in Chikwawa, Malawi. First, we present formal narratives of doing the 'right thing' in relation to accessing care and medicines, situating these within local and civic moralities. Second, we describe the types of antibiotics accessed and the local economies of accessing medicines, amid constrained possibilities for care and anticipation of inevitable future illness. Third, we explore practices of building up, applying, and sharing empirical knowledges about individual bodies and the medicines that work best for them. Thus, we highlight the complex sets of relations that underpin 'lay' logics for accessing medicines in a context of precarity and frequent illness.

### Being a responsible citizen

AN and ES were visiting households to conduct medicine interviews. As they arrived at one compound, they were greeted warmly met a lady who seemed to be in her mid-30s. Her house

was a simple structure with mud walls and a grass thatched roof. The household's main livelihood was subsistence farming growing maize, millet, and sorghum for food. Any extra grain they were able to grow was sold for cash, but this only happened in the years when rains were good. During the interview explained they were a household of eight and she had never attended school. She said that the family were frequently sick and needed medicines from the hospital. She explained that cotrimoxazole was the most frequently used drug in the household. She went on to say she only gets medicine from the hospital or buys what the doctor has prescribed. She used the example of her eleven-year-old son, who suffers from persistent stomach problems and had recently been given metronidazole at the hospital. When ES probed about whether drugs were ever stored in the house, she explained that they never store medicines in the house because the doctors always advised her not to. She also said that they never kept left over drugs because they always finished the full prescription. When asked where she would go next time, she said she would return to the hospital (Medicine interview ES and AN).

As can be seen in the fieldnote above, when discussing experiences of navigating illness and seeking care, participants often conveyed narratives of the 'right' thing to do in terms of accessing and taking medicines. Participants talked of the importance of accessing 'expert' care through public health facilities, presenting symptoms and illness in the appropriate way, and accessing and taking medicines according to formal prescriptions only.

Expressing the importance of seeking formal health care was a common feature of interviews about experiences of illness and treatment-seeking. When asked what they would do next time they get ill, many interview participants stated that they would "*rush to the hospital*" to seek care. This common narrative often referenced the availability of testing at formal facilities, leading to the prescription of "*appropriate treatment for the illness*" medicines. This reflects and reproduces official health messages about the role of tests in 'rationalising' the use of medicines [49].

Talk of doing the right thing 'next time' can be seen as a public performance of citizenship, recognising what is required to be deserving of care and recovery. It also reveals the disconnect between this normative intention and the contextual reality of navigating illness, [50] described in detail below, where the idea of 'rushing' to the hospital is an extremely challenging prospect for many, and neglects other, important temporalities of care-seeking in this context.

A second narrative of doing the 'right' thing as a citizen and patient centres on the importance of taking medicines in the 'right' way, specifically taking medicines only as prescribed and not accessing medicines privately unless instructed. While descriptions of saving and sharing medicines were common, these practices were often presented as something 'others' do: "*it isn't supposed to be happening*" (female farmer, residential interview, Njeleza). It was also framed by some in terms of the collective impact, reflecting a civic responsibility towards taking medicines in the 'right' way, to avoid potential harm to others:

> "*I think this behaviour contributes greatly to persistent cases of malaria amongst people in this community. This is because the dosage is stopped before malaria is completely treated*" (ES in-depth interview; residential ethnography, Mfera).

Furthermore, formal narratives around purchasing medicines privately, outside instruction by clinicians, involved criticisms of others, again setting apart those who recognise the importance of accessing medicines appropriately and those who do not. In the quote below a male participant during an informal group discussion narrated:

> '*people in the area have a tendency of buying antibiotics from local groceries. . . some people are just lazy and reluctant to take patients to the hospital*' [The statement was met with a

*general nodding and agreement from the group of men and women]* (AN fieldnotes; male farmer, residential ethnography, Njeleza).

Here, within a group setting there is an implication of a weak morality in the accusation of 'lazy' people who purchase medicines privately rather than seeking formal care. Verheijen's work (2018) highlighting conditionalities around networks of reciprocity among women in Malawi, whereby being perceived to be 'lazy' or wasteful of resources which may render someone undeserving of support, such as the sharing of medicines [51]. This suggests the importance of being seen to be doing the 'right' thing within local social networks. These enactments of moral personhood are required for being positioned favourably within networks of relations that are vital support systems in a context of extreme poverty and frequent ill-health.

The narratives can be seen as normative accounts, through which participants conveyed the 'right thing to do' around care-seeking, and thus constructed themselves as good citizens, often in contrast with others. These normative accounts resonate with the focus of global health campaigns around AMR, such as the WHO co-ordinated World Antibiotic Awareness Week (WAAW) [52]. People living in Chikwawa could articulate the 'rational' behaviours expected and promoted through such messaging, which seeks to educate lay publics and correct 'irrational' behaviours of overusing or misusing antibiotics, assuming they are rooted in a deficit of knowledge [53]. However, as we go on to illustrate, these public narratives do not reflect the reality of how decisions are made about accessing care in contexts of precarity, amid local financial, social, and moral economies, and in relation to personalised knowledge and experience of illness.

## Frequently accessed antibiotics and the local economies of accessing medicines

Across Chikwawa District, we found cotrimoxazole and amoxicillin to be the most frequently accessed and used antibiotics in all households we included (see Fig 1). Both antibiotics are on the access list according to the WHO AWaRe Classification, which means they should be easily accessible to patients when clinically needed [54].

We found that it was commonplace for residents in Chikwawa District to seek medicines beyond the formal public health sector. For many, this was an economic decision: accessing the 'free' state-provided health care was more costly than purchasing medicines from local providers, such as private pharmacies, grocery shops, market sellers or mobile drug vendors. Subsistence farming left households with few opportunities to access cash. Thus, decisions about forgoing income generation and other basic activities for survival, such as seeking food, by spending time seeking care were presented as challenging, in addition to the difficulties of finding money for transport to the clinic. This can be seen during a conversation AN held with a mother who had recently visited the clinic with her daughter. She described her decision to go to the hospital in the following way:

> *"I went to the hospital because she [my daughter] wasn't getting better, so I decided to take her to the hospital [the primary health care facility]. I borrowed MK 600 [approximately $0.74] from my neighbours. I spent some of the money on a bicycle taxi and the rest on a snack and drink because the wait at the hospital is so long." (Female resident and primary caregiver, AN; residential ethnography, Mfera)*

This scenario was compounded by the frequency of medicines being out of stock at the health facility. A familiar feature of health care in low resource settings,[55] the unpredictable,

## Antibiotics freqently used in Chikwawa

**Fig 1. Most frequently used antibiotics.**

and frequent stock outs of essential medicines in facilities in Malawi often leaves patients having to purchase prescribed medicines themselves [41]. In Chikwawa, there was a limited number of licenced pharmacies, and for most residents, local informal drug vendors or grocery stores were the most accessible to purchase medicines. Due to the limited number of medicines available in the Malawian health system, public facilities were considered to offer little more than informal providers. This can be seen in the quote below, during this conversation, AN

was talking to a subsistence farmer about why people did not go to the hospital and the farmer narrated:

> *"the drugs we receive at the hospital are the same drugs, we can buy them from the local grocery, so people consider it a waste of time to go to the hospital."* (Male farmer, Ethnographic fieldnotes AN, Residential ethnography, Njeleza).

This reflects how 'care' is commonly reduced to the provision of medicines [41,49]. If the formal health facility provides scarcely more (if any) 'care' than the local drug vendor, strategies for accessing medicines locally are highly rational when the costs of 'free' care are so high for daily survival. This reality is therefore incompatible with the assumed 'irrationality' of the behaviour to be 'corrected' through WAAW and other public health education campaigns around antibiotic 'misuse'.

The local circulation of medicines in Chikwawa District also involves anticipating future illness events, which prompts saving medicines for sharing and borrowing among friends, families, and neighbours. This reflects a temporality that is shaped by precarity, and the ongoing sense that misfortune will likely occur at any time [26,51]. Through the medicine interviews, participants narrated how frequently household members fell sick and the frequency with which they took antibiotics:

> "*we need this medicine [cotrimoxazole] almost every week*" (female farmer, Medicine interview AN Mfera).

In this context, keeping medicines from drugs dispensed at the public health clinic once a patient has recovered was considered a necessary act for preparing for and mitigating against the challenges of inevitable illness both within the household and the wider community. In the quote below, the female farmer described the practice of stopping medicine early to ensure they have some spare medicine:

> *"most women stop giving medicine to their children the moment their children start feeling better, so they keep the remaining medicine which is shared amongst the communities"* (Interview with female farmer, ES ethnographic fieldnotes Mfera).

While acknowledging that 'not finishing a prescribed dose' can be considered problematic regarding the recognised civic responsibility around antibiotics, the act of saving medicines is embedded in local networks of reciprocity that are often vital for people's access to care at "crucial timings"[51]. This highlights two intersecting dimensions of temporality experienced in this context of precarity. The everyday reality of illness requires urgent decisions about care that are set against other decisions for daily subsistence, as well as imaginings of the future, necessitating contribution to wider social relations that offer potential for support going forwards.

Sharing saved medicines locally, between friends, family, and neighbours, is formally acknowledged as something that is *"not supposed to"* happen. Yet we heard multiple accounts of people sharing leftover medicines with others. This was sometimes framed as a necessary act, in terms of the timing of an illness episode, occurring, for example, when access to health facilities or medicine providers was not possible. A child falling ill late at night may be considered particularly time-critical in terms of accessing medicine.

AN arrived at a compound in the morning. He was visiting a family after he was told that the family were asking for medicines during the night. When he arrived at the compound, he

met a lady whose daughter had been sick in the night. He asked her what had happened, and she explained that her daughter had a high fever and was breathing fast. She was alone in the compound as her husband was away and wanted to urgently get help for her daughter. She had no money for a taxi and no way of asking her husband for the money. She explained that she went to neighbours close by who she hoped would be able to give her medicines to help her daughter get better. She explained that one of the neighbours gave her Bactrim [cotrimoxazole] which she had then given the medicines to her child. In the morning, she felt her daughter was looking better because she had taken the medicines (Female farmer, Residential ethnography, AN, Njeleza).

This sense of urgency, necessitating the sharing of medicines, may also reflect wider global health messaging in malaria campaigns about the importance of early treatment-seeking for fever in children [56]. Talk of sharing medicines also revealed the contribution of the practice to established and ongoing social relations:

*"I share the medicines with my family and with a neighbour who is my sister. We know each other well and we always sincerely tell each other if we have medicine available in our houses or not."* (female farmer in-depth interview AN during ethnographic fieldwork, Mfera).

This example suggests a temporality of continuing support over time, constructed around the likelihood of medicine being available to share in the future, if not right now.

These reciprocal exchanges of medicines are part of a broader system of circulation of basic commodities, a necessary strategy for survival in a context of extreme poverty [51]. When the expected reciprocity does not happen, this is potentially damaging to the social relations around which it is built. It highlights again the unique temporalities around accessing medicine, where past exchanges shape expectations for care-seeking in the moment:

*"It felt painful to have been denied medicine because I usually help that friend with medicine anytime she asks me for help. It was like she took my help for granted"* (female resident in Mfera, AN, IDI during residential ethnography).

These quotations also suggest a gendered nature to forms of local care through the sharing of medicines. This was acknowledged in some participants' accounts of when and why sharing medicines happens locally, describing women as being "sympathetic" towards the needs of others, and therefore more likely to share. It may also reflect the dependency of women on local social relations for access to basic resources in Malawi, as described elsewhere [51]. These social and moral relations are situated within temporalities of access to medicines-as-care in the moment, and at any time. These tempos of antibiotic need are marginalised in AMR awareness campaigns, which rest on logics of acquired knowledge to be applied in a particular moment of deciding to access care.

## Medicine practices informed by local, empirical knowledges

Individuals' empirical understandings of their bodies, their illnesses and what medicines offer intersect with decisions around where and how to access medicines. This knowledge is established through multiple illness and care-seeking encounters and is applied in pragmatic ways to manage individual illness, and as part of local social and moral exchanges.

People's understanding of *their* illnesses as distinct are brought to light particularly in accounts of the limitations of formal care, which were experienced as incompatible with their individual bodies. As well as the costs of accessing formal health care, tests and medicines offered at facilities could conflict with personal expectations for how to treat illness. This is

evident in relation to testing for malaria through the malaria rapid diagnostic test (mRDT), and subsequent access (or otherwise) to lumefantrine/artemisinin (LA). Many participants' accounts reflected past frustrations that mRDTs failed to detect 'their' malaria, and therefore restrict their access to LA which is experienced as effective in alleviating their symptoms:

*"many people choose to buy LA from the shops at Dembo because they know that sometimes they are not diagnosed malaria positive due to their type of blood, so they simply choose to buy LA for them to get recovered."* (Male farmer, Residential ethnography, AN Mfera).

Experiences of this disparity led people to bypass formal care, applying their empirical knowledge of what works for *their* malaria and seeking medicines such as LA locally. This practice contributes to broader narratives of the failure of formal care to provide adequately for people's health needs, resulting in the necessity of other strategies to access the right medicines for their bodies.

Yet, past experiences of attending health facilities and receiving prescribed medicines do play an important role in applying knowledge of medicines in local contexts. Participants conveyed how knowledge gained through clinical encounters, as well as empirical experiences of illness and recovery, can be recalled and employed in decision-making around medicines, both for themselves and others in their social networks. In the quote below, a mother explains why she asks for LA from the neighbours:

*"Every time my child is sick, she is diagnosed with malaria and that is why I ask for LA from the neighbours, it always works and my child recovers very well"* (mother and resident in Mfera, AN in-depth interview, Mfera).

Similarly, this knowledge of medicines can be transferred to purchasing medicines from private providers where information on medicines and dosage may not be provided:

*AN was sat on the veranda chatting to a local male resident in his early thirties. AN was asking him about his experiences of buying medicines from an informal drug vendor. The man narrated that he often bought medicines locally as it was easier than visiting the clinic. He explained that the drug vendor didn't tell him how to take the medicine, but he knew from his previous experiences of using antibiotics (amoxicillin, cotrimoxazole and pen V) that he should take two tablets in the morning and evening. (Male resident of Njeleza, AN ethnographic fieldnotes)*

Knowledge acquired through multiple experiences of illness and recovery-seeking applied to fill the gaps in the form of care available through others reflects what Rodrigues (2020) describes as a form of 'legitimacy' in self-medication practices [57]. Previous formal prescriptions combine with empirical encounters of illness and recovery to contribute to "a more autonomous form of self-care" (ibid, p5) for (perceived) familiar conditions [57].

The building up and application of this knowledge also intersects networks of reciprocal relations around sharing medicines. Sharing often included description of the information about the recommended dosage. The passing on of advice about taking medicines, reflecting the transference of accumulated knowledge, is an important part of the exchange, and can be considered part of the act of care in itself. This practice highlights the social dimensions of rationales for seeking medicines. They are built not only on individualised "embodied experiences" of illness and recovery, but also on the experiences of others, and relations of trust and reciprocity within social networks [57].

## Discussion

Global awareness campaigns have been at the forefront of community interventions to address AMR [58]. To date, these campaigns have principally framed AMR as a problem of overuse with images and messages travelling across the world through coordinated international campaigns. In rural Chikwawa, precarity was an embodied experience for many. Accessing healthcare and medicines was challenging, and at times could place a significant economic burden on the household when care was sought. In this context of scarcity, the notion that antibiotics could be taken as unthinkingly as sweets, challenges the prevailing discourse that antibiotics are being overused everywhere. Residents articulated how performing the role of being a good citizen could be achieved through only seeking care and medicines from the formal health system reflecting normative messages often communicated by global health and development initiatives. Yet, these messages are heard, understood, and rearticulated, but are often incompatible with the lived reality of those households' dependent on subsistence farming for their survival. The performance of being a good citizen and helping neighbours through sharing drugs appeared also to be part of localised moralities governing relations of sharing and reciprocity which were incompatible with global health messages. Our findings suggest that it was not lack of knowledge but sociocultural and economic factors that shaped the way medicines and care were accessed in rural Chikwawa. This provides further evidence that the knowledge-deficit framing to address AMR is problematic, particularly in low-income contexts [22,59,60].

Anthropological studies have long established that the way people respond to and interpret ill-health rarely fit seamlessly within biomedical rationalities. In the field of pharmaceutical studies, social researchers have demonstrated that when seeking to understand medicine use, there is need to move beyond framing behaviour as either rational or irrational, to establish people's situated logics of medicine use [17,61]. In Maputo, Rodrigues's work on self-medication practices found that decisions to take medicines beyond (non-) prescription use were shaped by individuals' socio-economic position and the broader therapeutic landscape [50,57]. This reflects our findings, that medicine use is a relational process that is situated within specific economic, political, and social configurations. Furthermore, our findings correspond with anthropological interpretations of the complex temporalities that shape illness experiences, and care-seeking practices [62]. Concerns about future selves (and others), and the implications of past encounters, necessarily affect what is possible at the time of illness, particularly in a context of extremely limited care and other resources for living.

Critical global health scholars have argued that global public health projects can maintain health inequities and sustain neo-colonial configurations between the Global North and Global South through focusing on the health inequalities rather than engaging in their root causes [62,63]. A case study from the Ebola epidemic in West Africa brings to the fore the way in which global health projects and practitioners failed to engage in the social relations and complex power dynamics that sustained the epidemic [64]. Overlooking these dynamics can reinforce power asymmetries and can continue to cause harm to the communities they seek to help [62,63]. Awareness campaigns, with messages and images developed in the Global North and transported to low-income contexts such as Chikwawa ask those least able to change their medicine use practice to do so for the global good. Echoing this critical public health scholarship, we can see that awareness campaigns, by failing to acknowledge the social, economic, and political factors that shape access and use of medicines of people in the Global South these campaigns have the potential to cause harm.

The findings from this study have important implications for addressing antimicrobial resistance in low-income contexts. They speak to the need for policy actors to critically reflect

on the appropriateness of awareness campaigns that draw on knowledge-deficit model and exported to the Global South, particularly in communities where the economic survival is a day-to-day challenge for many households. If antibiotic awareness campaigns are to be used in these contexts, the messaging needs to be tied into broader economic development to ensure that basic needs are met, and that communities have basic levels of security to cope with future illness and economic shocks. In this way, efforts to address AMR can align with wider efforts to meet sustainable development goals (SDGs). Subsistence farming is likely to become more precarious as climatic change increases the frequency of flooding and extreme temperatures in Chikwawa, which is likely to further impact people's health and well-being. Reflecting these contextual factors into public health interventions such as awareness programmes will be vital.

## Limitations

In the Chikwawa district, the Malawi-Liverpool-Wellcome Trust Clinical Research Programme has a prominent and established presence in the region. Local communities readily recognise vehicles and researchers coming from the programme, and likely associate its work with a set of expectations around biomedical care and the duties of the participant. This may have shaped the presentation of public narratives of doing the right thing and should be interpreted in relation to the research context. Ideally the ethnographic fieldwork for the study would have run for longer than 6 months to allow for further deepening of relationships between the study team and residents in the two villages. However, due to the COVID-19 pandemic and the suspension of all field activities the study team stopped data collection. ES and AN have extensive experience of conducting health research in the area, which helped to contextualise the findings and supported the analysis helping to overcome some of this study limitation.

## Conclusion

Our paper demonstrates that people living in Chikwawa need better access to antibiotics and improved care. Focusing national and international attention on addressing this crisis in care rather than trying to stop people from taking antibiotics is likely to be a more fruitful and successful intervention.

## Supporting information

**S1 Text. Standards for reporting qualitative research.**
(DOCX)

**S2 Text. Household survey questionnaire.**
(DOCX)

## Acknowledgments

We are grateful to all study participants who took part in this study and to the FIEBRE and DRUM Consortiums.

## Author Contributions

**Conceptualization:** Eleanor E. MacPherson, Joanna Reynolds, Justin Dixon, Clare I. R. Chandler.

**Formal analysis:** Eleanor E. MacPherson, Joanna Reynolds, Esnart Sanudi, Alexander Nkaombe, John Mankhomwa, Justin Dixon, Clare I. R. Chandler.

**Funding acquisition:** Clare I. R. Chandler.

**Investigation:** Esnart Sanudi, Alexander Nkaombe, John Mankhomwa, Clare I. R. Chandler.

**Methodology:** Eleanor E. MacPherson, Esnart Sanudi, Justin Dixon, Clare I. R. Chandler.

**Writing – original draft:** Eleanor E. MacPherson, Joanna Reynolds, Justin Dixon, Clare I. R. Chandler.

**Writing – review & editing:** Eleanor E. MacPherson, Joanna Reynolds, Justin Dixon, Clare I. R. Chandler.

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
