## [Decision Letter · Decision Letter 0]

2 Dec 2021

PGPH-D-21-00742

Understanding antimicrobial use in subsistence farmers in Chikwawa District Malawi, implications for public awareness campaigns

Dear Dr. MacPherson,

Thank you for submitting your manuscript to PLOS Global Public Health. After careful consideration, we feel that it has merit but does not fully meet PLOS Global Public Health’s publication criteria as it currently stands. Therefore, we invite you to submit a revised version of the manuscript that addresses the points raised during the review process.

The feedback from the two reviewers provides a clear path for revising the manuscript to improve some unclear areas and to strengthen the reporting of the study. When the manuscript is returned following revisions, please provide the completed SRQR (Standards for reporting qualitative research) as a supplemental file with the page and paragraph numbers noted for each element (https://www.equator-network.org/reporting-guidelines/srqr/). In addition, I would encourage the authors to consider reviewing the COREQ (Consolidated criteria for reporting qualitative research) specific to the reporting recommendations for interviews (https://www.equator-network.org/reporting-guidelines/coreq/). Although returning this checklist is not required, the reporting elements should be addressed in the methods section. However, the checklist can also be completed by the authors with explanations about each reporting element within the document.

We look forward to receiving your revised manuscript.

Kind regards,

Patrick A. Palmieri, DHSc, DPhil(Hon), EdS, MBA, MSN, PGDip(Oxon), ACNP, RN, CPHRM, CPHQ, FAAN

Academic Editor

Journal Requirements:

1. Please include additional information regarding the survey or questionnaire used in the study and ensure that you have provided sufficient details that others could replicate the analyses. For instance, if you developed a questionnaire as part of this study and it is not under a copyright more restrictive than CC-BY, please include a copy, in both the original language and English, as Supporting Information.

2. Please provide separate figure files in .tif or .eps format only, and remove any figures embedded in your manuscript file.  If you are using LaTeX, you do not need to remove embedded figures.

3. In the online submission form, you indicated that your data will be submitted to a repository upon acceptance.  We strongly recommend all authors deposit their data before acceptance, as the process can be lengthy and hold up publication timelines. Please note that, though access restrictions are acceptable now, your entire data will need to be made freely accessible if your manuscript is accepted for publication. This policy applies to all data except where public deposition would breach compliance with the protocol approved by your research ethics board. If you are unable to adhere to our open data policy, please kindly revise your statement to explain your reasoning and we will seek the editor's input on an exemption. Please be assured that, once you have provided your new statement, the assessment of your exemption will not hold up the peer review process.

4. Please amend your detailed Financial Disclosure statement. This is published with the article, therefore should be completed in full sentences and contain the exact wording you wish to be published.

i) State the initials, alongside each funding source, of each author to receive each grant.

ii). State what role the funders took in the study. If the funders had no role in your study, please state: “The funders had no role in study design, data collection and analysis, decision to publish, or preparation of the manuscript.”

Reviewers' comments:

Reviewer's Responses to Questions

**Comments to the Author**

1. Does this manuscript meet PLOS Global Public Health’s publication criteria? Is the manuscript technically sound, and do the data support the conclusions? The manuscript must describe methodologically and ethically rigorous research with conclusions that are appropriately drawn based on the data presented.

Reviewer #1: Yes

Reviewer #2: Yes

2. Has the statistical analysis been performed appropriately and rigorously?

Reviewer #1: N/A

Reviewer #2: N/A

3. Have the authors made all data underlying the findings in their manuscript fully available (please refer to the Data Availability Statement at the start of the manuscript PDF file)?

Reviewer #1: No

Reviewer #2: Yes

4. Is the manuscript presented in an intelligible fashion and written in standard English?

Reviewer #1: Yes

Reviewer #2: Yes

5. Review Comments to the Author

Reviewer #1: This is an important paper on antimicrobial resistance drawing on empirical research to identify the life logics that shape the use and circulation of medicines in Malawi. The paper mobilises an anthropological approach, weaving together a combination of medicine interviews, ethnography and semi structured qualitative interviews to identify these dynamics.

The paper makes the case that, without addressing systems issues, awareness campaigns may fail to exert their intended effect and work instead to reinforce inequalities and fail to protect the efficacy of medicines.

Specific questions and comments and listed below:

• The authors described a public awareness image circulated in Malawi. How does circulation take place? Is it via whatsapp, or other means? Greater detail would be useful here.

• Subsistence farmers are also described as being targets of awareness raising campaigns – how and where does this targeting take place?

• Incorporating the image of the awareness campaign described in the introduction would help to bring the description to life.

• It felt at times that the authors could amplify further the significance of their findings by connecting their rich empirics to broader theory building efforts which demonstrate that interventions that are not in tune with context can not only fail to exert their intended effect but can also be accompanied by unintended consequences. Shove’s Beyond the ABC of Climate Change has sparked a lot of work in this regard and would be one of a number of relevant examples to connect to.

• Line 50-51 – missing word ‘of’

• Where is Malawi situated in relation to poverty indicators?

• The materials and methods would benefit from addressing the following questions:

• How long was data collection suspected during flooding?

• What did the day-to-day reach look like during the 6-month ethnography period – what did participant observation look like? How did the researchers keep notes during this time – were notes and observations written up daily?

• The methodology lacks specificity re the coding process (e.g. did you identify first and second order codes and was this done by some or all members of the research team?). How did you enable your empirics to breathe and not to be over-determined by theory?

• The analysis section unpacks three broadly defined sets of relations that shape how and why people seek medicines outside of formal health care. It would be useful to clarify which (if any) of the themes were more prominent than others. Were there any relations or logics/minor themes that did not meet a coding threshold?

• Labelling of field notes and interview extracts – not always consistent - it wasn’t always clear if quoted participant material was from the same or a different participant. Quotes of empirics are made e.g. line 254-255 without reference to e.g. interviewee number.

• The analysis is well written and provides rich anthropological narrative. However, the start of the ‘doing the right thing section’ moves very quickly from a fieldnote extract to a statement about multiple ‘normative accounts.’ Engaging more empirical material prior to referring to ‘these normative accounts’ would help the transition from empirical richness to the analysis narrative.

• The data is being made available via COMPASS but is not available yet and doesn’t accompany the submission.

Thank you for the opportunity to review this engaging contribution.

Reviewer #2: The manuscript is very interesting and well written. It will be good to see more information on the included households or health care professionals quotes (gender, age, education). The author can link the issues presented within this manuscript to sustainable development goals and argue on how people of Chikwawa will not be left behind? because sustainable development is connected to global health, this will enrich the discussion on what would be the needed interventions and basic foundation for policy maker in order to intervene.

6. PLOS authors have the option to publish the peer review history of their article (what does this mean?). If published, this will include your full peer review and any attached files.

**Do you want your identity to be public for this peer review?** For information about this choice, including consent withdrawal, please see our Privacy Policy.

Reviewer #1: No

Reviewer #2: **Yes: **Amani Eltayb

---

## [Editor Report · Decision Letter 1]

15 Feb 2022

PGPH-D-21-00742R1

Understanding antimicrobial use in subsistence farmers in Chikwawa District Malawi, implications for public awareness campaigns

Dear Dr. MacPherson,

Thank you for submitting your manuscript to PLOS Global Public Health. After careful consideration, we feel the revised manuscript requires a minor revision to meet PLOS Global Public Health’s publication. Therefore, we invite you to submit a revised version of the manuscript that addresses the final points raised during the review process.

We look forward to receiving your revised manuscript.

Kind regards,

Patrick Albert Palmieri, DHSc, EdS, MBA, MSN, PGDip(Oxon), RN, FAAN

Academic Editor

Journal Requirements:

Additional Editor Comments (if provided):

Thank you for submitting this revised manuscript. I believe the reviewer noted limitations and recommendations for improvements have been addressed. However, I have four methodological issues that need to be clarified by the authors. Please keep in mind I am well aware of the need to completed additional analysis with qualitative data. There are space limitations for manuscripts and multiple objectives in many larger studies, especially ethnographic and mixed methods. However, the manner in which this data is reported needs to be clearly, concisely, and transparently described. For the most part, the revised manuscript reads well but the four points below need to be further addressed, either through revisions or with clearly articulated arguments about why any of the revisions are not necessary. Again, the manuscript will quickly move forward with an acceptance once the next four points and a final recommendation are addressed. Thank you in advance for your patience and understanding.

FOUR POINTS

1 - Please clarify the study design. There is not a study design stated.

2 - Please clarify the study as reporting either an analysis from a primary (original) data collection or secondary use of previously collected data. After reviewing the published protocol, a third less likely option is the original study was segregated into different segments for reporting (but I am not sure after my review of all the documents).

3 - Please clarify the method of data analysis.

4 - Address the limitations inherent in this study design, especially if the data was a secondary analysis.

ADDITIONAL INFORMATION

1 - Yes, I am familair with critical medical anthropology, however this is not a study design. From what I understand, this was an ethnographic study, is this not correct? If correct, please state this to be the case at the beginning of the methods section with the appropriate methodological citations. If not correct, please justify the study design as written. Later in the methods section "anthropologically informed methods" is not appropriate to meaningfully address the question about study design.

"This research was conducted as part of the FIEBRE study, a multi-country and multidisciplinary investigation of febrile illness and antimicrobial use in five countries in Africa and Asia (43). The social science study in Malawi, explored the roles of antimicrobials in both primary health care provision and within the community and utilised an ethnographic methodology that was informed by critical medical anthropology (44)."

2 - The study seems to be a secondary data analysis resulting from a larger ethnographic study. If this is the case, please revise the methods to clearly state this to be the fact with the citation for the larger study, and the protocol.

"This paper reports primarily on the community-based research, but is informed by findings from research we conducted in the health care settings which are presented more fully elsewhere (45)."

3 - The data analysis is not clearly explained. What was the specific process for data analysis? The work done for the current study is not addressed. Was the analysis original from the secondary data? Was the data analyzed inductively, deductively, or perhaps both ways?

"Frequencies of recognition and use in households were established based on the structured medicine interviews. Analysis for the ethnographic fieldwork and in-depth interviews was conducted inductively through the course of the research. We used a collaborative approach to analysing the ethnographic and interview data. During the six-month data collection period, we held weekly debriefing sessions with the team to reflect on data collection progress and processes. During these meetings we identified and addressed any ethical concerns encountered in the field and identified emerging themes and any new insights or avenues for enquiry to be followed up in the next week. We also built on work initially undertaken within the primary health care facilities. The development of the first coding frame took place iteratively, building on the reading of the data, and the debriefing sessions. All transcripts and fieldnotes were imported into NVivo 12 and the programme was used to aid data analysis by coding against the thematic framework. Data was coded in NVivo after the data collection was completed, with two further analysis workshops held virtually with the whole research team to allow for and reflection throughout the process."

4 - The limitations section in the revised manuscript is not sufficiently descriptive about the limitations for this type of study. Please address the limitations in the study design, not merely the setting and context of the responses from participants.

"In the Chikwawa district, the Malawi-Liverpool-Wellcome Trust Clinical Research Programme has a prominent and established presence in the region. Local communities readily recognise vehicles and researchers coming from the programme, and likely associate its work with a set of expectations around biomedical care and the duties of the participant. This may have shaped the presentation of public narratives of doing the right thing and should be interpreted in relation to the research context."

RECOMMENDATION -----

Finally, I would like the authors to consider highlighting the aim of the previously published study:

"We highlight the contours of what we refer to as antibiotic vulnerabilities, exploring what constitutes care within primary health care facilities in Chikwawa District, in Malawi and how this relates to antimicrobial prescribing."

THEN, link this to the extension of this point in the aim of this study:

"The aim of our study was to provide an in-depth understanding of the ways medicines were accessed and used by households dependent on subsistence farming in the Chikwawa District of Malawi."

This would clearly explain this work resulted from a larger project and help readers to understand what is added by the current study.
---

## [Editor Report · Decision Letter 2]

9 Mar 2022

Understanding antimicrobial use in subsistence farmers in Chikwawa District Malawi, implications for public awareness campaigns

PGPH-D-21-00742R2

Dear Dr. MacPherson,

We are pleased to inform you that your manuscript 'Understanding antimicrobial use in subsistence farmers in Chikwawa District Malawi, implications for public awareness campaigns' has been provisionally accepted for publication in PLOS Global Public Health.

Best regards,

Patrick A. Palmieri, DHSc, DPhil(Hon), EdS, MBA, MSN, PGDip(Oxon), ACNP, RN, CPHRM, CPHQ, FFNMRCSI, FAAN

Academic Editor

Editor Comments (if any, and for reference): Thank you for addressing the areas requiring clarification.